# Phosphodiesterase Inhibition and Immunotropic Activity of Dipyridamole Dynamic Derivatives

**DOI:** 10.3390/cimb47040214

**Published:** 2025-03-21

**Authors:** Artur Martynov, Boris Farber, Alexander Katz

**Affiliations:** 1Laboratory and Clinical Department of Molecular Immunopharmacology, Mechnikov Institute of Microbiology and Immunology of National Academy of Medical Sciences of Ukraine, 61000 Kharkiv, Ukraine; 2R&D Department, TRIZ Biopharma Innovations LLC, 22 Perry Rd, Edison, NJ 08817, USA; drfarber@trizbiopharma.com; 3Department of Mathematics and Computer Science, St. John’s University, Queens, NY 11374, USA; katza@stjohns.edu

**Keywords:** dipyridamole combinatorial supramolecular derivatives, HPLC analysis, inhibition of phosphodiesterase in vitro, immunomodulation in vivo

## Abstract

*Introduction.* Many pharmacological properties of dipyridamole (DIP) are associated with its ability to inhibit phosphodiesterases (PDEs). Actually, DIP has interesting properties like antiviral for influenza, SARS-2 COVID-19, and herpesviruses. Our research aimed to design and synthesize the dynamic combinatorial DIP derivatives with more pronounced inhibiting properties in relation to PDE and to carry out the HPLC analysis of the resulting combinatorial derivatives of DIP. This study is aimed at investigating the effect of the dynamic derivative of dipyridamole (DDD) on intestinal dysbiosis syndrome in mice caused by streptomycin against the background of cyclophosphamide-induced cellular immunodeficiency. *Materials and methods*. For the synthesis of a dynamic combinatorial derivative of dipyridamole, we used a molecular dynamic method for drug design and combinatorial acylation of dipyridamole by succinic and acetic anhydride in different molar ranges of acylation agents. Combinatorial derivatives were analyzed using gradient HPLC with a UV detector. Also, derivatives established the inhibition ability for phosphodiesterase by the spectrophotometric method. Also, we used an in vivo mouse model with immunodeficiency caused by cyclophosphamide for pharmacological study. *Results and discussion*. Molecular modeling suggests that 18 different dipyridamole derivatives can self-assemble into a stable supramolecular structure with lower total energy. Specific combinatorial molar ratios of the synthesis components were necessary to create a new supramolecular compound with enhanced pharmacological properties. The inhibition of phosphodiesterase in such a dynamic combinatorial derivative already appeared at a concentration of 0.05 μM. In mice with colitis caused by streptomycin treatment, the administration of DDD per os resulted in an antidiarrheal effect and prevention of the animals’ weight loss. Given the cyclophosphamide-induced immunosuppression and streptomycin-associated diarrhea, immunity was completely restored only under the action of DDD. *Conclusions*. The most effective dipyridamole derivative for phosphodiesterase inhibition was formed only if the number of different derivatives in solution was maximum and consisted of all 18 molecules. With other quantities of modifiers, there was no qualitative change in the inhibitory activity of the combinatorial mixture against phosphodiesterase. According to all parameters, DDD has been proven to be more effective than the pure dipyridamole reference product.

## 1. Introduction

All modern medical products are static chemical structures, which has led to only slight statistical differences in efficacy compared with placebos [1]. Current static drugs are not functioning effectively, as almost 40% of people are not responding to them. This is because human receptor sites vary slightly from person to person, while a static drug always has the same fixed structure [2,3,4]. The effectiveness of a drug depends on whether it can bind to specific receptor sites in a person’s body. It is similar to finding the right fit between a hand (the drug) and a glove (the receptor). If the glove is too small for the hand, the drug will not be able to bind to the receptor and will not work. So, a drug may only be effective in certain individuals whose receptor sites are compatible with it [5].

For instance, let us consider the standard treatments available for hypertension. To regulate high blood pressure, doctors usually prescribe a combination of different medications, including antihypertensive, diuretic, and antiarrhythmic drugs. However, it is important to note that what works for one person may not be effective for another. Through polymorphism of the receptor and its gene, different variations can occur that may affect the receptor’s function. A hypotensive agent that successfully controls blood pressure for one individual may not have the same effect on another [6]. Previously, we have developed dynamic insulin [7], alpha-interferon [8], interleukin [9], and gamma-interferon [10], the efficacy and spectrum of action of which were significantly higher than the existing static drugs. Below is an example of insulin dynamization (Figure 1) [7].

We fragmented insulin using proteolytic enzymes and modified amino acid residues by replacing some with carboxyl groups to create smaller fragments with modified charges. We created a mixture of thousands of insulin fragments, allowing only the matching fragments to bind to the receptors of a specific patient [11]. These types of insulin fragments can be easily taken orally in the form of tablets. This is possible because the insulin is broken down into fragments by proteolytic enzymes, and these fragments are protected from further destruction by acylation, which alters the charges on them. For this project, chimeric insulin was used and administered orally to rats with alloxan-induced diabetes, which yielded impressive results. According to our research, the level of blood glucose in rats reduced drastically from an average of 40 mmol/L to 9 mmol/L after a single dose of dynamic insulin. This dynamic insulin dose was enough to maintain the glucose level in rats at 9–11 mmol/L for 24 h. In comparison, the control group of rats had a glucose level ranging from 38 to 42 mmol/L. Based on these findings, it may be possible to develop dynamic insulin tablets that can be taken once a day for patients with type 1 diabetes and those with insulin resistance type 2 diabetes [7].

Dipyridamole is a very interesting substance due to its ability to inhibit several phosphodiesterases at once, including PDE 5–11 [12]. It has a wide range of biological activities such as direct antiviral and interferon induction, anticancer, anticoagulation, antianginal, wound healing, stem cell division stimulation, angiogenesis, and many more [13,14,15,16,17,18,19,20]. The biological properties of the most common phosphodiesterase inhibitors are reviewed [21].

The concept of drug dynamization involves using a blend of structurally similar drugs to create a superstructure that can fit any lock or target receptor. This is different from using a static drug that only works for one receptor. In the case of modified dipyridamole, at least 5 new compounds are formed when it is modified with succinic anhydride, and 18 new derivatives are formed when simultaneously modified with two modifiers. These derivatives can form a metastable superstructure that is constantly changing and adapting to external environmental conditions, the human body, and target enzymes such as phosphodiesterases. One of the top 50 most frequently prescribed medications is dipyridamole (2,6-bis-diethanolamino-4,8-dipiperidinopyrimido(5,4-d)-pyrimidine), a pyrimidopyrimidine derivative [22]. It is possible that variables other than platelet inhibition play a role in the drug’s ability to prevent thrombosis in experimental mice. Additionally, Dip may aid tissue regeneration by promoting cAMP accumulation [23]. In addition to these direct actions, dipyridamole may also enhance aspirin’s ability to suppress platelets through a pharmacokinetic interaction [14].

Dip also proved to be active against influenza viruses [24]. The antiviral activity measured using different techniques ranged from 90 to 99 percent. In tissue cultures, three dipyridamole derivatives were considerably effective against the influenza A/England 42/72/A/FPV viruses. Dipyridamole revealed a 62.5 percent protection rate when given orally to white mice infected with influenza virus (A/England 42/72) [25].

It has been demonstrated that Dip, a safe medication with good and extensive pharmacological capabilities, prevents EBV reactivation from B-cell lines. DIP repurposing could soon be studied, either alone or in combination with other antivirals, to treat EBV-related disorders where lytic replication plays a harmful role because of its extensive clinical use [26].

Herpes simplex virus thymidine kinase has been demonstrated to be important for reactivation from latency. Also, the effect of Dip on HSV reactivation has been studied [19].

Earlier, it was found that Dip markedly potentiates the antiviral effects of the derivatives of adenine and 2, S-diaminopurine and 3-deazaadenine [27]. The 50% inhibitory concentration of the acyclic nucleoside phosphonates for VZV-, HSV-, and HCMV-induced cytopathic effect or plaque formation significantly decreased as a result. Dip did not enhance the activity of vidarabine, acyclovir, or ganciclovir. These results were confirmed by virus yield assays (for HSV and HCMV) and flow cytometry (for VZV) [27].

Dip also induced interferon in mice after intravenous administration. Peak interferon levels in the blood (128 IU/mL) were attained 49 h after injection of 0.1 mg dipyridamole per kg body weight and at 24 and 12 h after injection of 0.6–1.8 and 16.7 mg/kg, respectively [28]. A single application of the interferon inducer dipyridamole administered intraperitoneally or per os demonstrated a marked antiviral effect in mice with experimentally induced alphavirus infection—Semliki Forest (SFV) virus-caused encephalitis, applied orally in influenza virus A/Puerto Rico/8/34 (H1N1) and B/Lee/40—respiratory infections, and herpes simplex virus type 1, strains Lennett and Leningrad 2 encephalitis, and HSV-1 strain Leningrad 2-caused skin infection [15]. This interferon-inducing capacity of dipyridamole may account for its broad-spectrum antiviral effect [29,30].

Aerosols containing dipyridamole exhibit strong reparative effects and considerably speed up the rat model’s epithelialization and wound healing. They perform this type of effect somewhat better than dexpanthenol. Aerosols containing dipyridamole have the power to stimulate the growth of pluripotent CD34 cells, which in turn has a positive impact on the immune system throughout the body. Papaverine and dipyridamole’s combined effect on tissues selectively stimulates the division of pluripotent cells in the wound and speeds up the recovery of the animal’s immune system after induced immunodeficiency by a factor of six.

Many of the above presented pharmacological properties of Dip are associated with its ability to inhibit phosphodiesterases (PDEs) of different classes [12].

Also, review [13] provided evidence advocating DIP as a possible therapy against major COVID-19 complications such as acute kidney injury, acute respiratory distress syndrome, and acute liver injury, encompassing the pieces of evidence directly from DIP.

The purpose of the research was to design and synthesize the combinatorial Dip derivatives with more pronounced PDE-inhibiting properties and immunotropic activity in vivo to carry out the HPLC analysis of the resulting combinatorial Dip derivatives.

## 2. Materials and Methods

### 2.1. Planning (Methodology) of Research

Thus, dipyridamole is a prospective basis for obtaining new derivatives with both antiviral and antiaggregant properties. We proposed a new method for obtaining dynamic supramolecular structures with the incomplete combinatorial substitution of available groups in the acylation reaction of hydroxyl residues (-OH) at the same time as two modifiers ((**II**) and (**III**) in Figure 2). Such structures should be considered not as a mixture of individual compounds of combinatorial synthesis but as a superstructure dynamically interacting with each other’s compounds and exhibiting significantly more pronounced pharmacological effects. In this manuscript, we provide synthesis, analysis, and in vitro activity of the obtained superstructure based on dipyridamole derivatives.

Figure 3 shows the principle of the dynamic structure formation with the calculated modification of dipyridamole with only one modifier (**III**), succinic anhydride. In the acylation reaction of dipyridamole, succinic anhydride replaces the residues of the -OH groups indicated on the left in Figure 3. Dipyridamole (**I**) is coded as D in Figure 3, and its conditional scheme is shown below and also signed as D. Due to a symmetry of the dipyridamole structure, its monosubstituted derivative mono succinyl dipyridamole (1SD) is represented by only one molecule and is shown in the diagram as part of a dynamic combinatorial structure. The top right of Figure 3 shows both the chemical structure of succinyl dipyridamole and its schematic representation labeled 1SD. Accordingly, the disubstituted derivative will be present in two different forms, 2SDa and 2SDb, and one trisubstituted derivative 3SD and one fully substituted derivative 4SD. These structures do not only exist in the usual mixture as separate molecules but also form more complex supramolecular structures with each other with a much more pronounced or altered pharmacological activity. When using two modifiers—acetic (**II**) and succinic (**III**) anhydrides—the number of components of the supramolecular dynamic structure increases from 6 to 18. To obtain a combinatorial mixture from many different derivatives, the number of reaction components is calculated using combinatorics Formulas (1) and (2). To test the hypothesis (Figure 3) that it is the superstructure with the maximum number of various substituted derivatives that has the maximum biological activity, we also obtained derivatives with a different molar ratio—(**I**):(**II**):(**III**). Figure 4 describes the entire methodology of the experiments performed.

### 2.2. Molecular Dynamics

In order to understand how the individual fragments of a combinatorial mixture come together to form a superstructure, modeling of the simultaneous interaction of 18 components of dipyridamole derivatives was necessary. Unfortunately, docking methods only allow for deterministic modeling of the interaction between a ligand and a receptor, making it impossible to simultaneously model the behavior and interaction of several substances. Only the water–cell molecular dynamics method offers the opportunity to do so. Although this process is slow and can take several weeks even on powerful hardware, it makes it possible to track the formation of a supramolecular complex of combinatorial derivatives in the model by allowing dozens of molecules to simultaneously interact with each other. The compound dipyridamole (I) has a double symmetry, which results in some hydroxyl groups being available for acylation in the combinatorial synthesis reaction. This yields identical molecules that can form more complex superstructures with each other. For instance, the reaction yield produces 18 different derivatives when two modifiers are used. Our previous studies have shown that these derivatives can interact with one another to form more complex structures. To investigate this further, we conducted molecular simulations of the behavior of all 18 dipyridamole derivatives with different substituents at the hydroxyl groups. The simulations were carried out in a 68.3 A water box (simulation cell).

A YASARA structure was used for molecular modeling, structure refinement, molecular dynamics simulations (MDs), and result presentation through molecular graphics [31,32,33,34] (license N 179465823, http://www.yasara.org/, accessed on 10 March 2025 ). The coordinates of the dipyridamole molecule were retrieved from the NMR experiment [35]. Each dipyridamole derivative was geometrically optimized in the MMFF94 force field and by the AM1 method [36]. The simulation involved placing molecular systems in a cubic periodic cell filled with TIP3P water molecules. The cell was 1 nm larger than the molecular system studied in all three dimensions. Prior to the simulation, the molecular systems were energy-minimized using a short steepest descent minimization followed by simulated annealing minimization. The MD simulations were run in the NPT ensemble at 298 K and pH 7.4, and included dynamic dipyridamole mixture interactions in a water solution. Na^+^ and Cl^−^ counterions were added to neutralize the systems and to reach an ion mass fraction of 0.9% NaCl. The AMBER14IPQ force field was used for the simulations [37], with a 1.05 nm force cutoff for dispersion interactions [38]. Long-range electrostatic interactions were calculated by the particle mesh Ewald algorithm [39]. The hydrogen atoms’ movements were limited in some way. The equations that describe their movement were calculated using a 2.5 fs step. To accelerate the process of calculation, the non-bonded van der Waals and electrostatic forces were only evaluated every second step and were adjusted with a scaling factor of 2 [31]. Trajectories were computed for 30 ns, and the data were saved each 25 ps.

The simulation modeled the interaction of components for 181 ns using simulation cells ranging from 82.7 to 82.9 A radius of gyration of the solute (water solution).

### 2.3. Synthesis of (IV)

The procedure first involved diluting 18 μM of dipyridamole (**I**) (CAS N 58-32-2, Sigma, St. Louis, MO, USA, Mr = 504.636 g/mol, n = 4) in 50 mL of dioxane in a mixture with 50 mL of glacial acetic acid, and adding 25 μM of succinic anhydride (**III**) and 25 μM of acetic anhydride (**II**). Then, the solution was stirred and warmed with a backflow condenser for 25 min. Then, the solution was poured into vials and lyophilized to remove solvents and acetic acid to create the combinatorial mixture (**IV**). The combinatorial mixture (**IV**) was used to make pharmaceutical compositions, to study structures, and to determine the bioactivity of (**IV)**. Figure 1 provides the scheme of combinatorial dipyridamole derivative (**IV**) synthesis. Figure 3 depicts one parent molecule of dipyridamole **(I)** that provided 4 residual hydroxyl groups that could be available for modification (n = 4) to create the combinatorial mixture (**IV**). The amino groups, as part of residual hexahydropyridine and the pyrimidine nucleus, can be protonated and protected against modification under the given reaction conditions (acidic medium). Calculations of the number of modifier moles were carried out according to the combinatorics formulas [40].

Thus, having only one parent dipyridamole molecule and two modifiers after combinatorial synthesis, we obtained 18 combinatorial derivatives with different degrees of OH- substitution, different positions of the substituents, and different shuffling of the modifier residues. This was not a simple mixture, but a difficult-to-separate supramolecular mixture. Due to the presence of both substituted and non-substituted hydroxyl groups in the different derivatives, the supramolecular structures were formed through both hydrogen and ionic bonds, including tertiary amino groups of heterocycles.

HPLC was performed using an HPLC microcolumn chromatograph (Millichrom A-02, Econova, Novosibirsk, Russia ) with a gradient of acetonitrile (5–100%)/0.1 chloric acid and 0.5 lithium perchlorate).

Similarly, as for (**IV**) (above), derivatives with other ratios (**I**) and modifiers (**II**, **III**) were prepared. The ratios of (**I**), (**II**), (**III**), and PDE inhibition activity.

### 2.4. Detection of Derivative (IV) Effect on Activity of PDE

For the detection of PDE activity, an ab139460 PDE activity colorimetric assay kit (Abcam, Waltham, MA, USA) was used according to instructions [41,42]. For fixing results, a plate spectrophotometer StatFax 303 (Awareness Technology, Palm City, FL, USA) was used at A = 620 nm, with a 10% measuring error. Native dipyridamole (**I**) was used as the positive control.

### 2.5. Animals, Diarrhea, and Immunodeficiency

Animal research was carried out in accordance with the European Convention for the Protection of Vertebrate Animals Used for Experimental and Other Scientific Purposes [43,44]. The bioethical aspects of the investigations were approved in accordance with international standards by the Committee on Bioethics of the I. Mechnikov Institute of Microbiology and Immunology of the National Academy of Medical Sciences of Ukraine State Institution [45,46]. Male mice weighing 24 to 25 g from a non-inbred laboratory strain were used and housed in typical laboratory settings (20–22 °C, 14 h/10 h light/dark cycle, 65% humidity) with free access to food and water. A balanced diet was included in the regular mouse feeding pellets. The practices used were consistent with the institutional requirements.

Streptomycin (STM) (Arterium, Kyiv, Ukraine, serial No. 125894) was delivered intragastrically to mice for 9 days to model the dysbiosis syndrome [47,48].

Immune status disorders were simulated by a single subcutaneous injection of cyclophosphamide (CFA) (Endoxan, Baxter oncology GmbH, Halle, Germany, Serial No. 4J035F) at a dose of 250 mg/kg. The CFA subcutaneous injection ensured the development of a long-term (7–10 days) immunocompromised state in mice [49]. The mice were administered CFA 1 h before the streptomycin single-dose administration.

D3 (line 3) was utilized at a dosage of 5 mg/kg. D3 was given to the animals intragastrically at a dose of 5 mg/kg. Pure chemicals were suspended in distilled water prior to the administration of D3.

The control medication was derivative 10 (D10 or line 10), which contained solely pure dipyridamole (Sigma, USA). D10 was administered at a dose of 5 mg/kg per os. Both drugs were given for 7 days after a 5-day antibiotic regimen. The aforementioned experimental technique was based on prior research that has shown that, in the case of streptomycin-induced pathology, feces liquefaction (diarrhea) was observed as early as the sixth day of drug administration.

Animals were randomized into 7 groups, each comprising 12 mice, as follows:Control (intact);Mice with diarrhea caused by STM;Mice with diarrhea caused by STM and immunosuppression caused by CFA;Mice with diarrhea caused by STM and treated by D3;Mice with diarrhea caused by STM and immunosuppression caused by CFA and treated by D3;Mice with diarrhea caused by STM and treated by D10;Mice with diarrhea caused by STM and immunosuppression caused by CFA and treated by D10.

Regimes of streptomycin, cyclophosphamide, and experimental probiotics applications in all experiments are shown in Table 1.

Animals, weighed prior to and on the 12th day of the experiment, were kept in cages in individual compartments.

#### Immunological Methods

*NO and cytokine production assay*. Murine peritoneal macrophages and splenic T-lymphocytes were isolated and TNF production was measured using a sandwich enzyme immunoassay (ELISA) approach with the ELISA kit (Mouse TNF-a ELISA Kit, PharMingen, BD Biosciences, San Jose, CA, USA), as described in [50]. TNF- is mostly produced by cells of the mononuclear phagocytic lineage macrophages; in contrast, TNF-b lymphotoxin is primarily produced by T lymphocytes.

TNF-a and TNF-b have highly similar biological characteristics, chemical structures (many antigenic epitopes are identical), and receptors. Both forms of TNF can be tested using the PharMingen system. TNF-b is detected when T-lymphocyte cells are cultured, but TNF-a is detected when macrophage cells are cultured. Macrophages were grown at 37 °C in a humidified atmosphere containing 5% CO_2_ in 24-well stripped plates (BioRad, Hannover, Germany) in Medium 199 with a methyl red pH indicator. The medium contained 25mM HEPES and 3% fetal bovine serum. At 21 h, supernatants were collected and tested for NO_2_. The Griess reagent was used to quantify NO_2_, which functioned as an indication of NO generation (1% sulfanilamide/0.1% naphthylethylene diamine dihydrochloride and 2.5% H_3_PO_4_ (1:1)). Following cultivation, 100 mL of each sample was deposited in a 96-well plate along with 100 mL of the Griess reagent. After ten minutes, the optical density was measured at 590 nm using a trip reader (Stat Fax 303 plus, Awareness Technologies Inc., Palm City, FL, USA) [51]. All ELISA studies utilized the same reader.

### 2.6. Preparation of Peritoneal Macrophages

Peritoneal macrophages from mice were isolated from a mixed white-cell population acquired via peritoneal lavage. After trypan blue and eosin exclusion, cells were washed three times with serum-free RPMI 1640 (Sigma, USA) supplemented with 2 mM glutamine (Merck, Darmstadt, Germany), 100 g/mL streptomycin, and 5 × 10^−5^ M-mercaptoethanol (Sigma, USA) and counted. Peritoneal cells were planted and allowed to adhere in flat-bottomed culture dishes (Costar, Washington, DC, USA). After plating for 1 h at 37 °C in a humidified atmosphere containing 5% CO_2_ in air, the cultures were thoroughly washed with jets of media from a Pasteur pipette to eliminate nonadherent cells. The remaining cells were 95% macrophages and formed a confluent monolayer 0.5 × 10^6^/mL. An inverted microscope was used to analyze the morphology of a monolayer. In RPMI-1640 medium containing 10% fetal calf serum, 1% of 1 M HEPES solution (Serva, Heidelberg, Germany), 2 mM glutamine (Merck), and 0.5% gentamicin (Sigma, USA), the incubation was maintained for 24 h at 37 °C. After incubation, the cell-free supernatant was collected and kept at 200 °C until TNF activity was determined. In certain tests, the cell suspension was frozen and thawed three times following incubation. TNF in this lysate was both cell-associated and secreted.

### 2.7. Preparation of Splenic T Lymphocytes

Spleens were aseptically removed and processed in glass homogenizers in 199 medium with 1% HEPES solution, 0.5% gentamicin, and 10% bovine serum (BS). Erythrocytes were lysed with Tris-buffered ammonium chloride (0.01 M Tris-HCl with 0.15 M NaCl-0.83% NH_4_Cl, 9:1) and washed thrice. The cells were then resuspended in RPMI-1640 media with 1% 1 M HEPES buffer, 0.5% gentamicin, and 5% fetal calf serum FCS to a final concentration of 7 × 10^6^ cell/mL. Exclusion of trypan blue and eosin determined cell viability. Panning was used to separate the populations of B and T lymphocytes in the spleen [52]. In 7 mL of phosphate buffer saline (PBS), plastic cell culture flasks 200 cm^3^ (Agilent, Santa Clara, CA, USA) were coated with affinity-purified rabbit immunoglobulin (Ig), (Sigma, USA) with specificity for rat Ig. After overnight incubation at 40 °C, the flasks were washed three times with medium and allowed to stand for at least 30 min in RPMI 1640 media. The flasks were then filled with 7 × 10^6^ spleen cells in RPMI-1640 media in 7 to 8 mL aliquots. Flasks were incubated at 40 °C for 60 min, with gentle stirring every 30 min to redistribute cells. Nonadherent cells were gently drained out, washed once, and resuspended in RPMI-1640 media containing 1% 1 M HEPES buffer, 0.5% gentamicin, and 2 mM and 2 mM L-glutamine, 5 × 10^−5^.

### 2.8. IL-2 Assay

The 1.5 × 10^6^ cells/mL spleen lymphocytes were cultured for 72 h on 24-well stripe plates in 199 medium at 37 °C in a humidified atmosphere containing 5% CO_2_. Then, 25 mM HEPES, 5% fetal bovine serum, and 5 mg/mL phytohemagglutinin (FHA) were added to the media. ELISA was used to assess the concentration of IL-2 in the supernatant of PHA-stimulated cells using rabbit antimurine IL-2 polyclonal antibody (BioRad, AbD Serotec, Raleigh, NC, USA) and goat antirabbit IgG biotin conjugate (Agrisera Antibodies, Vännäs, Sweden). The optical density was measured on the strip reader at 405 nm after staining with ABTS [53].

### 2.9. Statistical Analysis

All the experimental data were processed by the method of variation statistics. Calculation of statistical significance in the case of nominal variables was performed using ANOVA and ANOVA for experiments with repeated measures. The number of animals in each group (n) was 12, sufficient for statistically reliable results (the minimum can be 5). Statistical significance was calculated by Student’s *t*-test in Bonferroni modification [54]. LibreOffice Calk 25.2 was used for the calculations.

## 3. Results

### 3.1. Molecular Dynamics

As can be seen from the result of molecular gyration modeling (Figure 5), the rotation radius varied in a rather significant range from 5.25 A to 6.20 A. At that, by 180 nanoseconds, the system stabilized in the region of 5.55 A–6.10 A.

It was only possible to measure the total RMSD (Figure 6) of a supramolecular structure consisting of 18 independent molecules that were not bonded with each other, when investigated through molecular dynamics in a water/salt medium. However, when the same supramolecular structure was input into the docking method after optimization, or, conversely, when docking was first attempted, program errors occurred. Currently, docking systems are not capable of multiligand simultaneous docking with several ligands simultaneously, or changing dynamic supramolecular structures based on individual but complementary compounds.

According to Figure 6, when all 18 components of the system were kept separately, the root mean square deviation (RMSD) gradually increased from 0.5 A to 4.5 A. Then, the molecules started forming complex supramolecular structures with two–three molecules clustering together, between 50 and 120 nanoseconds. The RMSD values for several small supramolecular structures ranged from 1.7 A to 3.5 A. When these small supramolecular structures merged into a single system, the RMSD fluctuations were narrower, ranging from 2.0 A to 3.0 A. This indicates that the system stabilized in the form of a supramolecular structure, which included all components of the system (as shown in Figure 7 and Figure 8).

Figure 7 and Figure 8 indicate that all 18 structures were partially complementary to one another, resulting in a chimeric superstructure. The properties of this superstructure are yet to be studied, but it differed significantly from both the original dipyridamole and all 18 individual dipyridamole derivatives. It is likely to have unique properties that are separately distinct from each synthesis product. The structure is more similar to a protein or nucleic acid (DNA or RNA).

### 3.2. Synthesis and Analysis of Dynamic Dipyridamole

The complex supramolecular structures formed between the 18 different combinatorial derivatives of the (**IV**) were not able to be fully chromatographically separated by this HPLC column method, but showed seven bands with the same UV spectra (Figure 9). In fact, at least seven compounds with a dipyridamole core and different molecular masses were synthesized. Initial dipyridamole (**I**) was indicated in Peak 6 (Figure 10).

Only seven components were separated at HPLC; some of them were more hydrophilic (Peaks 2, 3, and 4, Figure 11) and exited from the column faster than the initial dipyridamole (**I**) (Peak 6 in Figure 10), while other derivatives in small quantities became more hydrophobic (Peaks 5–8 in Figure 11). All these peaks had the same UV spectra and contained the same pyrimidine dipyridamole core. As a rule, such combinatorial derivatives form more complex supramolecular structures with each other, which are not separated by HPLC. Many minor derivatives were masked by other absorption peaks, because their numbers were slight (Figure 11). Due to the fact that the main molecular mechanism of dipyridamole’s action is to inhibit phosphodiesterases 5–8, 10, 11 (PDEs), the structure/activity dependence of a number of dipyridamole’s combinatorial derivatives with a different ratio to the modifiers was investigated.

### 3.3. Study of the Derivative’s Phosphodiesterase Inhibition Ability

Next, a study was carried out of the cAMP-phosphodiesterase inhibition by different combinatorial dipyridamole derivatives (including (**IV**), item no. 3 in Table 2) obtained in the reaction with different molar ratios of modifiers (k1 and k2) according to the final concentration of AMP by the ELISA method. The reaction was stopped by the addition of a double volume of 1% 3-chloroacetic acid (TCA).

**Table 2 cimb-47-00214-t002:** Inhibiting properties regarding cAMP-phosphodiesterase (PDE) from supramolecular combinatorial derivatives of dipyridamole (IV, item no. 3) and derivatives of (I) obtained in the reaction with different molar ratios of modifiers (II) and (III).

Item No.	Reagent Molecular Ratio *			IC_50_ **^1^, µM
	m	k1	k2	
1	18	100 ***	100 ***	>800
2	-//-	50	50	160
3 (D3)	-//-	25	25	0.05
4	-//-	15	15	40
5	-//-	7	7	40
6	-//-	3	3	20
7	-//-	1	1	20
8	-//-	0.5	0.5	10
9	-//-	0.25	0.25	10
10 (D10)	-	0	0	5
11	Control from Kit 3-isobutyl-1-methylxanthine (IBMX)			5

* m—number of moles of dipyridamole in the combinatorial synthesis reaction; k1—number of moles of succinic anhydride in the reaction; k2—number of moles of acetic anhydride in the reaction; ** IC_50_ μM for derivatives of (I) at PDE inhibition (two range dilution); *** excessive amount of modifiers in relation to -OH groups in dipyridamole are replaced, past of modifiers in the reaction are unreacted, succinic anhydride and acetic anhydride remain in medium; ^1^—measuring error 10%.

As Table 2 shows, the smallest IC_50_ was observed precisely in the region with the calculated molar ratios of the modifiers (18:25:25). Thus, due to obtaining the supramolecular combinatorial derivative of dipyridamole (**IV**), the effective dose of the drug can be reduced 100 times to completely inhibit PDE.

Derivatives (N 2 and N4), nearly increased by ratio modifiers, inhibited PDE only at high values of 40 μM and 160 μM. At the same time, a smooth increased ratio in k1 and k2 was observed for all derivatives, which was inversely proportional to the activity decreases for more part derivatives, except the supramolecular derivative (IV) only. This fact indirectly confirms the formation of a more complex supramolecular system with significantly changed biological activity.

### 3.4. In Vivo Results

The advancement of diarrhea had a negative impact on the weight of the animals (Table 3). The weight of mice in group 1 (intact) grew by 11%, whereas in groups 2 and 3 (pathology and immunosuppression), the weight decreased by up to 15% and 12.2%, respectively.

Streptomycin at high dosages is known to promote inflammatory bowel disease (colitis) in mice and to reduce their resistance to infection [22]. Weight loss was caused by the advancement of diarrhea, parietal digestion issues, and absorption disorder. The administration of D3 against a background of pathology aided in the improvement of parameters and, as a result, weight gain.

### 3.5. The Effect of D3 and D10 (See Table 2) on Mice Immunity

The effect of streptomycin on the mice immune system was not prominent (Table 4), and was accompanied by a significant reduction in IL-2 production.

TNF production was dramatically reduced by macrophages and T-cells, while NO generation was boosted by macrophages. The number of splenic lymphocytes and macrophages reduced dramatically, although not below the life-threatening level for the animals. The combination of CFA and STM had a considerable impact on both the worsening of intestinal dysbiosis and the immune system (group 3); T-lymphocytes produced more than five times as much IL-2 in the spleen and blood plasma. TNF-a levels reduced in a similar manner. Also, while TNF-b levels were much lower, they were not as important as in group 1. The NO generated by macrophages reduced about two-fold in group 3, which included animals given cyclophosphamide and streptomycin. Among the experimental groups, the highest results for immunological recovery to near-normal levels were seen in group 5, where D3 was delivered against a CFA background. The control group had larger levels of splenic T-lymphocytes (although not statistically significant, *p* < 0.1). The parameters in the animal groups that did not receive artificial immunosuppression (groups 4 and 6) were similar to the control group 2 (streptomycin). However, only the values from group 4 (D3 introduction on the background of STM) differed substantially (*p* < 0.05), and all parameters were comparable to those from control group 2 (streptomycin). The only incomprehensible phenomenon that went beyond the concept of immunosuppression was that IL-2 generated by splenic lymphocytes exceeded the control values in group 5 (with induced immunosuppression). This was due to the non-uniform action of CFA on distinct sections of the immune system, along with their non-synchronous stimulation by D3. In contrast with D3, D10 stimulated immunity compared with control group 3, but it could not recover, even to the level of control group 2 (with streptomycin).

## 4. Discussion

According to our research, we found 18 different combinations of derivatives of dipyridamole (IV) that can result in complex supramolecular structures. Using the HPLC column method, we were unable to fully separate these structures, which showed seven bands with the same UV spectra. However, we were able to synthesize at least seven compounds with a dipyridamole core and different molecular masses. Typically, combinatorial derivatives like these form more complex supramolecular structures that cannot be separated by HPLC. By obtaining a supramolecular combinatorial derivative of dipyridamole (IV)—specifically D3 and D10—we can significantly reduce the effective dose of the drug (up to 100 times) to completely inhibit PDE. Furthermore, we observed a smooth increase in the ratio of k1 and k2 for all derivatives, except for the supramolecular derivative D3 (IV), which showed a significantly changed biological activity. This indirectly confirms the formation of a more complex supramolecular system. We propose that the supramolecular system is more stable in the enzyme’s active center. At a dose 100 times lower than the original dipyridamole (D10), it inhibits enzyme activity. Therefore, we can predict that the D3 derivative will exhibit a higher degree of antiviral activity and a higher potential for inducing interferon.

It is widely accepted that high dosages of streptomycin can cause inflammatory bowel disease in mice and weaken their ability to fight infections. This leads to weight loss, diarrhea, digestive problems, and difficulty in absorbing nutrients. However, administering D3 supplements to mice with these symptoms has been shown to improve their condition and promote weight gain. Streptomycin has been found to significantly reduce the production of IL-2, an important protein that helps regulate the immune system. It also reduces the production of TNF by macrophages and T-cells, while increasing the generation of NO by macrophages. As a result of taking streptomycin, the number of splenic lymphocytes and macrophages in mice decreases significantly, but not to a life-threatening level. Accordingly, a combination of CFA and STM has a considerable impact on both the worsening of intestinal dysbiosis and the immune system (group 3). As we have shown above, D3 was administered along with a CFA background. The control group had higher levels of splenic T-lymphocytes, although the difference was not statistically significant. However, the values from group D3’s introduction to the background of STM were significantly different, and all parameters were similar to those from control group 2. The only phenomenon that went beyond the concept of immunosuppression was that IL-2 generated by splenic lymphocytes exceeded the control values in groups with induced immunosuppression. The results obtained suggest a promising new avenue for the research and application of dynamic dipyridamole derivatives. Further investigations of D3 are necessary to explore this possibility. The stability of the supramolecular complex in both liquid and dried states is not well understood. It is known that dried supramolecular structures can lose their original biological activity, similar to hydrosols and other aqueous suspensions, which cannot be resuspended to restore their original biological activity.

The obtained results indicate that the dynamic derivative of dipyridamole is promising and, accordingly, it is necessary to conduct further studies on the manifestation of antiviral properties by this structure, taking into account the presence of such properties in dipyridamole. It is also planned to study the dynamic derivative of dipyridamole as an interferon inducer. Due to the fact that dynamic structures in the form of supramolecular mixtures have not been previously studied for biological activity and effects on immunity, in the future, it will also be necessary to develop a methodology for the standardization of supramolecular dynamic preparations, taking into account the multiplicity of composition.

## 5. Conclusions

Only certain combinatorial ratios of the synthesis components (k1 and k2 for (II) and (III)) make it possible to obtain a new supramolecular compound with more pronounced pharmacological properties. The inhibition of phosphodiesterase in such a dynamic combinatorial derivative already appears at a concentration 0.05 μM (on the pyrimidine core recalculation of dipyridamole). Such a compound is formed only if the number of different dipyridamole derivatives in the solution is maximum and consists of all 18 molecules. With other quantities of modifiers, there is no qualitative change in the inhibitory activity of the combinatorial mixture against phosphodiesterase.

In mice with streptomycin-induced colitis, administration of dynamic dipyridamole (D3) resulted in an antidiarrheal effect (diarrhea syndrome was reduced by 75%), normalization of gastrointestinal motility, and reduction of weight loss. The antidiarrheal effect was less evident in the case of cyclophosphamide-induced immunosuppression, implying that the immune system plays a part in the mechanism of action (PDA inhibition). D3 normalized the immune system’s quantitative parameters (the quantity of splenic lymphocytes, macrophages, and T-lymphocytes) and the cell functional activity (IL-2 and TNF production). Given the cyclophosphamide-induced immunosuppression and streptomycin-associated diarrhea, only dynamic dipyridamole totally restored immunity.

Thus, D3 was proven to be more effective than the pure dipyridamole (D10) reference product across all criteria.

In the short term, research into D3’s antiviral, anticancer, and interferon-inducing properties is on the horizon.

## Figures and Tables

**Figure 1 cimb-47-00214-f001:**
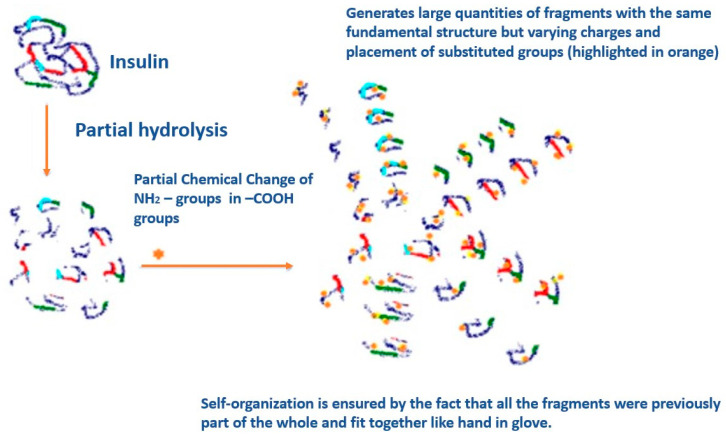
Mechanism of insulin dynamization and the formation of chimeric dynamic self-assembled insulin.

**Figure 2 cimb-47-00214-f002:**
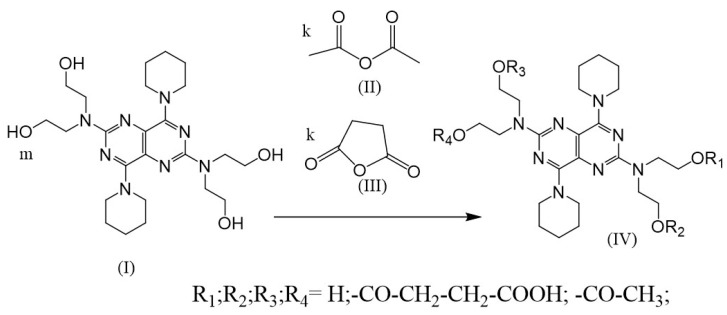
Scheme of combinatorial dipyridamole derivative (IV) synthesis, which comprises a combinatorial reaction of dipyridamole (I) with two modifiers (II, III). After the reaction, 18 distinct derivatives (IV) are formed that perfectly complement each other. The reaction is carried out in a mixture of dioxane and acetic acid, and the reaction mixture is boiled with a reflux condenser for 25 min.

**Figure 3 cimb-47-00214-f003:**
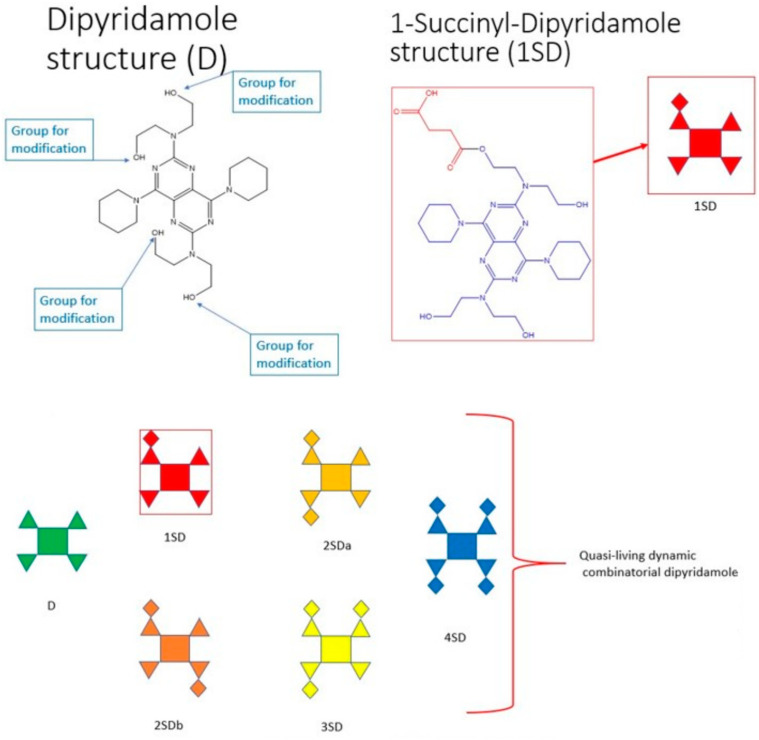
Self-assembled dynamic combinatorial derivatives of dipyridamole and principal of their synthesis (to simplify, the scheme is shown for ONE modifier, succinic anhydride).

**Figure 4 cimb-47-00214-f004:**
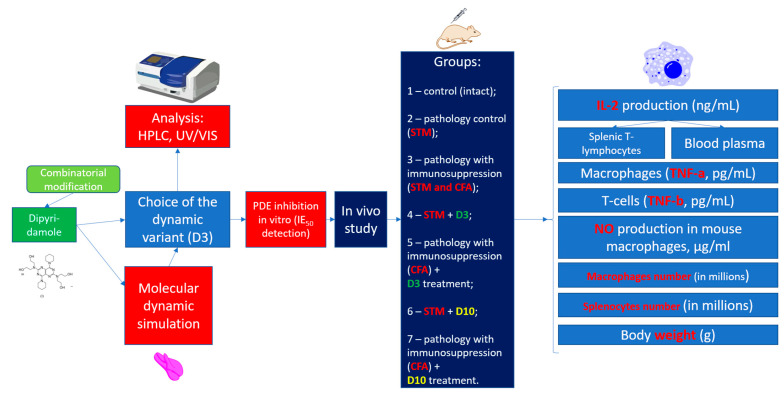
Methodology of the experiment.

**Figure 5 cimb-47-00214-f005:**
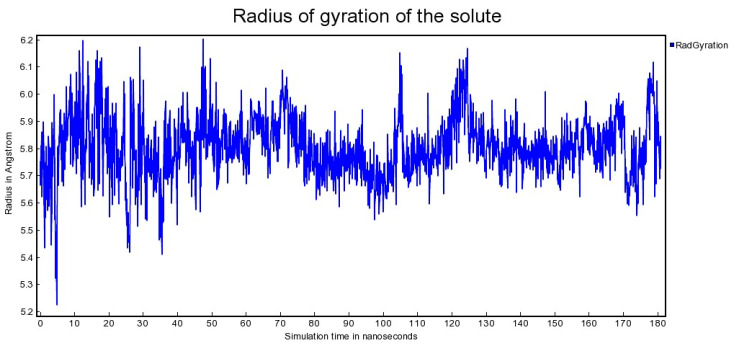
Radius of gyration of the solute [vertical axis] as a function of simulation time [horizontal axis].

**Figure 6 cimb-47-00214-f006:**
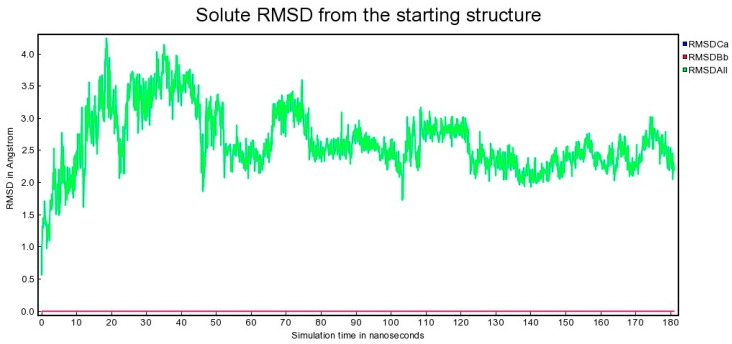
Solute RMSD from the starting structure [vertical axis] as a function of simulation time [horizontal axis]. Note: Graph RMSDCa, graph RMSDBb have all zero values. Graph RMSDBb completely covers graph RMSDCa; they share the same values.

**Figure 7 cimb-47-00214-f007:**
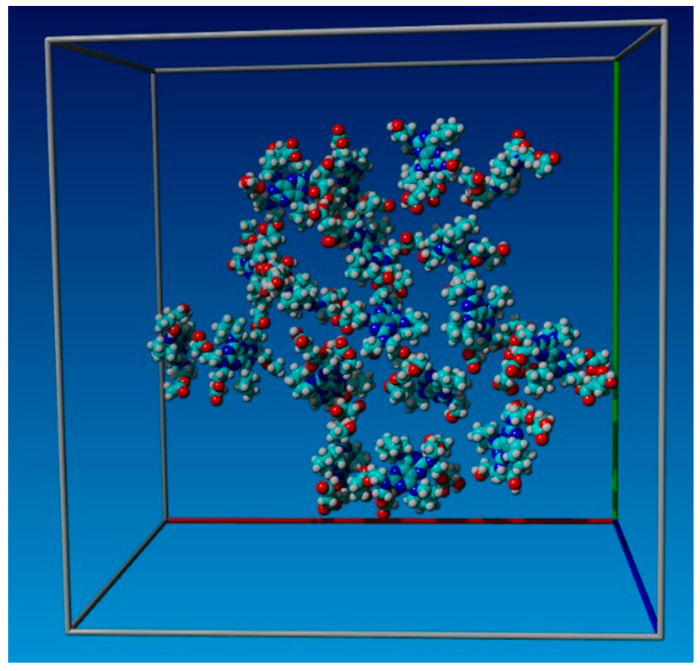
Combinatorial modification of dipyridamole with succinic and acetic anhydride produced an initial mixture of 18 derivatives with varying affinities that complement each other.

**Figure 8 cimb-47-00214-f008:**
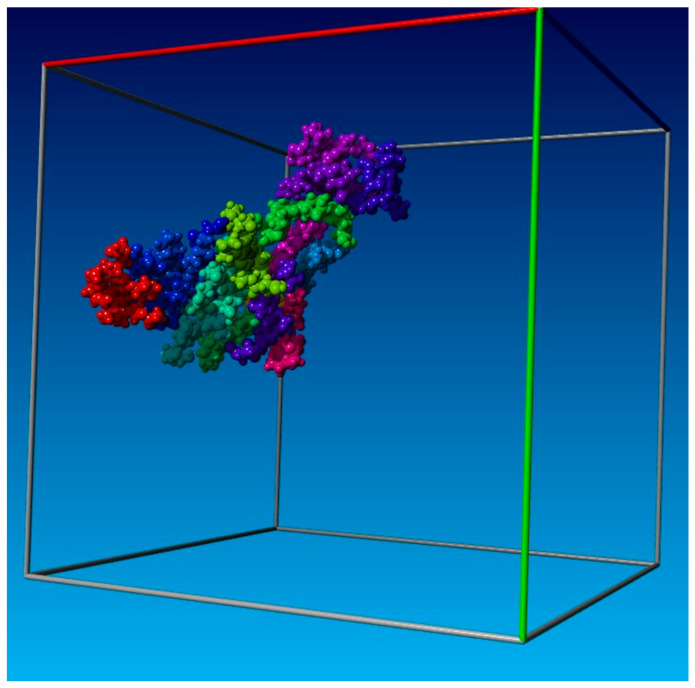
During 180 ns of AMBER force field molecular dynamics simulations, 18 dipyridamole derivatives form a single stable supramolecular structure. Each derivative is colored for ease of perception.

**Figure 9 cimb-47-00214-f009:**
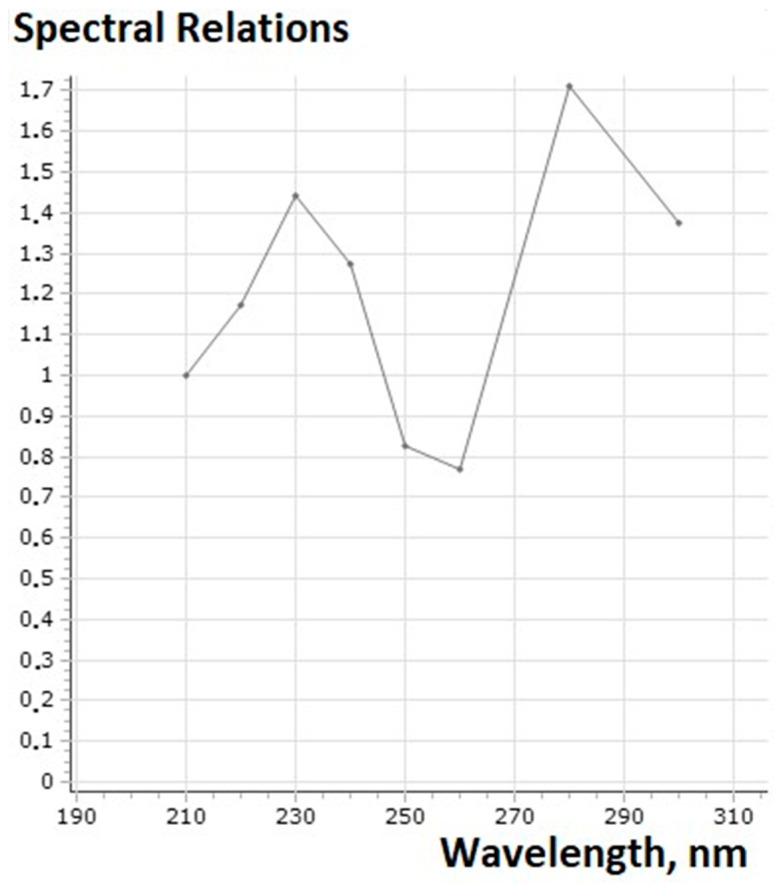
UV-spectral relation (Xnm/210nm) (HPLC Millichrom A-02, UV-detector) similar for all peaks (N 6 in Figure 3 and N 2–8 in Figure 4).

**Figure 10 cimb-47-00214-f010:**
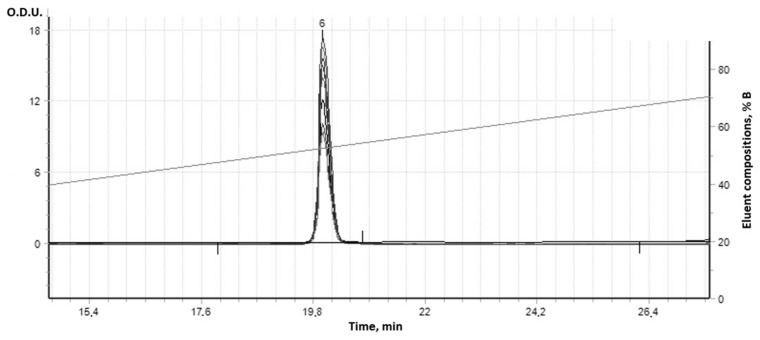
HPLC chromatogram of dipyridamole native (I) in gradient AcCN/aqueous solution LiClO_4_/HClO_4_ buffer.

**Figure 11 cimb-47-00214-f011:**
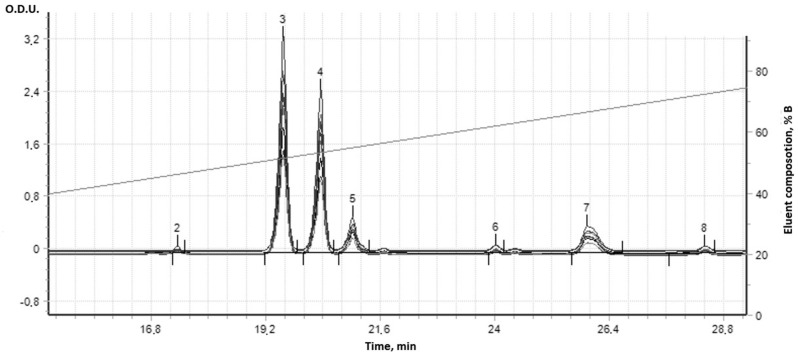
HPLC chromatogram of dipyridamole combinatorial derivative (18 substances in supramolecular complexes) (IV) in gradient AcCN/aqueous solution LiClO_4_/HClO_4_ buffer.

**Table 1 cimb-47-00214-t001:** Regimes for substance applications in animals.

Used Substances	Day of Experiment											
	1	2	3	4	5	6	7	8	9	10	11	12
Streptomycin	+	+	+	+	+	+ *	+ *	+ *	+ *	*	*	*
Cyclophosphamide	+ **	**	**	**	**	**	**	**	**	**	**	**
D3 and D10						+	+	+	+	+	+	+

+—application substances in animals on the days of this experiment; *—diarrhea in animals after using streptomycin; **—immunosuppression in animals after using cyclophosphamide.

**Table 3 cimb-47-00214-t003:** Effect of the streptomycin-induced diarrhea and DD on mice weight.

Mice Group (n = 12)	Body Weight (g)	
	Before Experiment	On the 12th Day
1	20.80 ± 0.52	23.12 ± 0.58 *
2	22.00 ± 0.28	18.60 ±0.45 *
3	20.88 ± 0.40	18.34 ± 0.66 *
4	20.82 ± 0.56	22.88 ± 0.82 *
5	20.70 ± 0.56	21.10 ± 0.50
6	20.65 ± 0.40	21.85 ± 0.40 *
7	20.78 ± 0.88	22.65 ± 0.28

*—statistically significant difference compared with the weight before the experiment (*p* ≤ 0.05).

**Table 4 cimb-47-00214-t004:** Effect of D3 and D10 on mice immunity.

Mice Group ** (n = 12)	Cells, Tissue, Lymphokines						
	Il-2 Production (ng/mL)		Macrophages (TNF-a) (pg/mL)	T Cells (TNF-b) (pg/mL)	NO Production in Mouse Macrophages, µg/mL	Macrophages Number (in Millions)	Splenocytes Number (in Millions)
	Splenic T-Lymphocytes	Blood Plasma					
1 control	73.92 ± 2.39	161.75 ± 5.15	15.33 ± 3.03	12.75 ± 2.01	9.92 ± 2.39	1.92 ± 0.67	127.00 ± 10.01
2 control	53.75 ± 4.73	107.42 ± 5.74	8.58 ± 1.98	10.83 ± 3.56	17.17 ± 3.74	1.33 ± 0.49	107.92 ± 10.67
3 control	11.75 ± 4.39	31.33 ± 4.31	2.58 ± 1.0	8.25 ± 1.14	6.33 ± 1.97	1.08 ± 0.29	89.00 ± 8.30
4	65.25 ± 5.29	155.50 ± 4.64	10.92 ± 1.98	10.17 ± 4.00 *	12.25 ± 2.01	1.83 ± 0.94 *	116.33 ± 5.25
5	86.33 ±5.18	160.00 ± 12.45 *	18.17 ± 1.34	14.25 ± 3.22 *	13.08 ± 2.23	1.42 ± 0.51 *	137.50 ± 10.88
6	51.33 ± 4.79 *	100.92 ± 8.13	7.25 ± 2.26 *	7.92 ± 2.11	10.08 ± 3.12	1.50 ± 0.52 *	105.50 ± 5.89 *
7	36.50 ± 4.34	79.25 ± 5.64	2.92 ± 1.44 *	4.17 ± 1.47	6.00 ± 1.41 *	1.25 ± 0.45 *	79.67 ± 8.40

* no difference compared with the controls, *p* > 0.05; ** group 1 (intact) was used as a control for group 5; group 2 was used as a control for groups 4 and 6; group 3 was used as a control for groups 5 and 7.

## Data Availability

This manuscript has data included as electronic Appendix A.

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
