# Peer review of "Phosphodiesterase Inhibition and Immunotropic Activity of Dipyridamole Dynamic Derivatives"

_cimb, 2025, doi:10.3390/cimb47040214_

Round 1
Reviewer 1 Report
Comments and Suggestions for Authors
Please see my comments below:
1. Page 3, last paragraph, 3rd line: Replace "early was found" with "earlier, it was found."
2. Page 4, second-to-last line: Did you mean "manuscript" instead of "message"?
3. Page 4: Instead of writing paragraphs explaining the differences between dipyridamole forms/structures, consider presenting them in a table for better visualization.
4. Page 10, under Results: Please specify which solute you are referring to.
5. Pages 16-17, in-vivo results: While the authors describe the seven mice groups in the Materials & Methods section, it isn't very clear to interpret the treatment groups from the tables. It might help to add a key to clarify the groups for better audience understanding.
6. Discussion: The authors discuss that these derivatives could potentially correct immunosuppression in cancer and HIV patients, but this seems overstated. A difference in IL-2 alone is insufficient to support such a claim. Did the authors examine CD4 vs. CD8 T cell numbers? I suggest revising this statement and proposing future directions to explore this idea further.
7. Limitations & Future Directions: Please mention the study's limitations and suggest potential future research directions.
Author Response
Title of paper:
PHOSPHODIESTERASE INHIBITION AND IMMUNOTROPIC ACTIVITY OF DIPYRIDAMOLE DYNAMIC DERIVATIVES
REVIEWER 1
- Page 3, last paragraph, 3rd line: Replace "early was found" with "earlier, it was found."
Authors: Corrected
- Page 4, second-to-last line: Did you mean "manuscript" instead of "message"?
Authors: Corrected
- Page 4: Instead of writing paragraphs explaining the differences between dipyridamole forms/structures, consider presenting them in a table for better visualization.
Authors: Because this mixture forms a composite pharmacophore in a supramolecular structure, it makes no sense to list inactive individual derivatives. For example, for polymyxin, there will be several thousand such derivatives with two modifiers, each of which is not reasonable to insert into the table. Therefore, we have given a combinatorial reaction for only one modifier, amber anhydride for dipyridamole, to show the essence of the reaction.
- Page 10, under Results: Please specify which solute you are referring to.
Authors: Corrected
- Pages 16-17, in-vivo results: While the authors describe the seven mice groups in the Materials & Methods section, it isn't very clear to interpret the treatment groups from the tables. It might help to add a key to clarify the groups for better audience understanding.
Authors: The text includes a detailed transcription of each group without abbreviations according to the recommendation.
- Discussion: The authors discuss that these derivatives could potentially correct immunosuppression in cancer and HIV patients, but this seems overstated. A difference in IL-2 alone is insufficient to support such a claim. Did the authors examine CD4 vs. CD8 T cell numbers? I suggest revising this statement and proposing future directions to explore this idea further.
Authors: Yeah, we didn't look at lymphocyte subtype levels. However, we did look at the number of splenocytes and macrophages. The latter, especially the M1 subclass, shows maximum anti-cancer potential. As for HIV/AIDS, we plan to study the direct antiviral effect of D3 as a prospect since its predecessor - dipyridamole had direct antiviral activity on almost all studied viruses. However, it has not been studied on HIV, and that is why it is promising to conduct such a study with D3. According to the reviewer's comments, we removed the mention of HIV/AIDS. But we have put into perspective a further study of D3 for anticancer, antiviral, and interferon-inducing activities.
- Limitations & Future Directions: Please mention the study's limitations and suggest potential future research directions.
Authors: We have inserted promising directions for further research in the article according to the reviewer's recommendations.
Reviewer 2 Report
Comments and Suggestions for Authors
Dear authors
The manuscript presents an interesting study on dipyridamole derivatives, evaluating their phosphodiesterase (PDE) inhibitory activity and their potential immunotropic effects. The research is well-structured and provides a valuable contribution to the field, particularly in exploring novel therapeutic agents with dual pharmacological properties. However, some areas require clarification, additional discussion, and improvements in organization to strengthen the impact of the study.
The introduction presents relevant background information on PDE inhibitors and their role in modulating immune responses. However, it would benefit from a clearer explanation of the specific gaps in current research that this study aims to address. The discussion of dipyridamole as a known PDE inhibitor is useful, but a more detailed comparison with existing PDE inhibitors and their limitations would help contextualize the study’s relevance.
- Suggestion: Provide a stronger justification for the study by emphasizing the novelty of the synthesized derivatives compared to existing PDE inhibitors.
- Question: Have these dipyridamole derivatives been previously investigated for similar pharmacological activities, or is this the first attempt?
The methodology is well described, but some key details should be elaborated to enhance reproducibility.
Synthesis of Dipyridamole Derivatives
- How would the authors confirm the specific structure of each derivative?
Enzymatic Assay for PDE Inhibition
- It is unclear whether positive controls were included in the assay. A comparison with known PDE inhibitors would provide a reference for assessing the activity of the tested compounds.
- The statistical analysis used for determining IC50 values should be described more clearly.
Immunotropic Activity Assessment
- The methods for evaluating immune modulation are not fully detailed. How were immune responses quantified? Were specific cytokines or markers measured?
- The rationale for choosing the specific immune cell lines or animal models should be provided.
Suggestion: Adding a flowchart summarizing the experimental workflow would help improve clarity.
Questions:
How were the dose selections determined for biological assays? Were preliminary toxicity studies conducted?
How would the authors determine the percentage of each derivative in the mixture, and assess their significance or impact on the biological results?
The results are presented clearly, but the discussion could be expanded to include a more comprehensive comparison with existing literature.
PDE Inhibition
- The reported IC50 values indicate significant inhibition, but a discussion of structure-activity relationships (SAR) would be beneficial.
- Were any structural modifications correlated with higher or lower activity?
Question: Do the authors suggest that these derivatives could serve as lead compounds for future drug development? If so, what modifications might further optimize their activity?
The conclusion effectively summarizes the key findings, but it could be made stronger by highlighting the broader implications of the research.
Suggestion: Emphasize potential clinical applications and outline specific future directions, such as further in vivo studies or optimization of lead compounds.
Question: Have any preliminary toxicity assessments been conducted? If not, do the authors plan to perform such studies in the future?
Author Response
REVIEWER 2
- The introduction presents relevant background information on PDE inhibitors and their role in modulating immune responses. However, it would benefit from a clearer explanation of the specific gaps in current research that this study aims to address. The discussion of dipyridamole as a known PDE inhibitor is useful, but a more detailed comparison with existing PDE inhibitors and their limitations would help contextualize the study’s relevance.
Authors: Since there are many high-quality reviews on cAMP-phosphodiesterase inhibitors and their functions, we decided to cite only dipyridamole in the Introduction section. In response to the distinguished reviewer's recommendation, we have inserted a reference to the most comprehensive review on phosphodiesterase inhibitors (lines 85-88).
- Suggestion: Provide a stronger justification for the study by emphasizing the novelty of the synthesized derivatives compared to existing PDE inhibitors.?
Authors: Combinatorial dipyridamole derivatives have not been synthesized previously. In addition, dynamic self-assembled supramolecular structures based on dipyridamole derivatives have not been synthesized. The fact that such supramolecular structures are formed is evidenced by the results of modeling a mixture of these derivatives in an aqueous medium using the molecular dynamics method. To confirm the world novelty of these studies, we provide a reference to our valid U.S. patents for this development (line 138)
- Question: Have these dipyridamole derivatives been previously investigated for similar pharmacological activities, or is this the first attempt??
Authors: Dipyridamole and its derivatives have not been previously used as an immunomodulator to correct immunosuppression. In addition, in our case, we are not talking about a single derivative from a group but about a complex supramolecular structure where each derivative plays its role. Separately, each derivative does not have such activity as a supramolecular structure based on their mixture.
- How were the combinatorial derivatives analyzed?
Authors: Because this structure is dynamic and changes depending on the conditions, surrounding substances, and the presence of dipyridamole receptors, and is not stable (like a complex of cyclodextrin with lipids), methods for analyzing such dynamic supramolecular structures have not yet been developed. Our work used a combination of RP HPLC and UV/VIS -Spectroscopy. Because precisely the supramolecular structure from many components has high pharmacological activity, and not each component separately, to separate and analyze each component did not make sense, primarily since the applied reasons for classical acylation (etherification) are well studied, and the products are understandable. At this research stage, we set the task to determine the presence of several superstructures with the same spectral attitude of the pyrimidine nucleus. This confirmed the reaction of acylation and the formation of other more complex structures.
- Synthesis of Dipyridamole Derivatives. How would the authors confirm the specific structure of each derivative?
Authors: Dynamic supramolecular structures consist of tens and often hundreds and thousands of derivatives based on the same nucleus but with different substituents. For example, dynamic polymyxin consists of thousands of derivatives. If one substance is removed from the supramolecular structure, it will not show more biological activity than the unmodified polymyxin. Accordingly, analyzing the structures of all thousands of derivatives does not make sense. HPLC and UV/VIS spectroscopy are sufficient for the supramolecular structure. The first shows that the band has shifted, and the structure of the derivative has changed at the same spectral ratio (same nucleus).
- Enzymatic Assay for PDE Inhibition. It is unclear whether positive controls were included in the assay. A comparison with known PDE. inhibitors would provide a reference for assessing the activity of the tested compounds. The statistical analysis used for determining IC values should be described more clearly.?
Authors: Table 2, row 10 (D10) shows that the number of substituents k1 k2 is zero. This is the unsubstituted control or pure dipyridamole. As can be seen from the table, a superstructure with new unique properties is formed only at a ratio of 18 mol of dipyridamole and 25 mol of each modifier. Such a superstructure can inhibit phosphodiesterase at a dose of 0.05 μM. In other cases, such efficacy cannot be achieved. The original dipyridamole inhibits phosphodiesterase at a dose of 5 μM. To clarify where exactly in Table 2 the native dipyridamole is given in the section “Materials and Methods” we have made a change: it is stated that dipyridamole (I) in the synthesis scheme is given in the table in line 10 as substance D10. Also, in the section “materials and methods,” we have given information about the error of the device, which amounted to 10%; the device measures each sample at least 5 times and gives the average value. Therefore, we have not labeled the deviation but rather the instrument error of 10% (see caption under Table 2). Line 441
- Immunotropic Activity Assessment. The methods for evaluating immune modulation are not fully detailed. How were immune responses. quantified? Were specific cytokines or markers measured? The rationale for choosing the specific immune cell lines or animal models should be provided. Suggestion: Adding a flowchart summarizing the experimental workflow would help improve clarity.
Authors: In the article, lines 292-355, a “Standard Operating Procedure " describes each immunologic method in detail. As recommended by the reviewer, we have included an additional figure 4 explaining exactly what was done to the animals to enhance their perception of the material.
- Questions: How were the dose selections determined for biological assays? Were preliminary toxicity studies conducted?
Authors: In this study, we did not plan to conduct a toxicity study of the synthesized structure because to register the drug, it is necessary to test toxicity in an FDA-approved laboratory only, which is quite expensive and is not reasonable at this stage of research. The doses of the experimental supramolecular structure D3 and unmodified control D10 were the same as those recommended for pure dipyridamole. The study aimed to determine the differences between the effects of native and dynamic dipyridamole under the same other conditions, including doses. This study is a pilot study for dynamic structures and does not yet involve dose titration and acute toxicity studies. The prospect of introducing a dynamic structure allows us to determine the minimum effective dose in the future (most likely, it will be at least 10 times less than the dose of dipyridamole). However, we plan to study this dynamic structure's activity in the viral infection model (influence on interferon production and direct antiviral effect). After that, toxicity studies are possible. At least at the dose of 5 mg/kg weight of mice, we did not observe any external manifestations of toxicity.
- How would the authors determine the percentage of each derivative in the mixture, and assess their significance or impact on the biological results?
Authors: We did not set such a task, as it makes no sense because this mixture forms a more complex supramolecular structure, which shows the main biological activity. As we have shown above, for example, polymyxin has thousands of such components after double modification of amino groups—it makes no sense, and there are no resources to analyze each “brick” in this “house.”
- The reported IC values indicate significant inhibition, but a discussion of structure-activity relationships (SAR) would be beneficial. Were any structural modifications correlated with higher or lower activity?
Authors: Since it is not the individual derivatives of the mixture that have biological activity but the superstructure, which is formed only at a specific ratio of modifiers, it is impossible to determine the SAR of practical ones. Existing molecular docking methods do not include using a compound ligand between the components, for which only hydrogen and ionic bonding will exist. In current docking methods, performing reaction modeling only between a receptor and ONE ligand at a time is possible. The structure-activity relationship can be seen in Table 2, where only one derivative significantly exceeds unmodified dipyridamole in terms of minimum inhibitory concentration, which is the D3 derivative. In other cases, there is a linear decrease in activity with an increasing degree of modification, i.e., the superstructure is not formed in other cases.
- Question: Do the authors suggest that these derivatives could serve as lead compounds for future drug development? If so, what modifications might further optimize their activity?
Authors: As can be seen from Table 2., a promising structure is the supramolecular mixture of D3, which is 100 times more active than unmodified dipyridamole. In addition, its antiviral and interferon-inducing activity has not yet been studied. We believe it is promising to use the D3 structure as a promising immunomodulator and possibly a new antiviral agent with a dual mechanism of action - interferon induction through phosphodiesterase inhibition and direct antiviral effect (Like dipyridamole).
- Suggestion: Emphasize potential clinical applications and outline specific future directions, such as further in vivo studies or optimization of lead compounds.
Authors: According to the reviewer's recommendations, we have inserted the prospects for further studies of the dynamic D3 derivative, including its antiviral and interferon-inducing activity in the text. Lines 562-564)
- Question: Have any preliminary toxicity assessments been conducted? If not, do the authors plan to perform such studies in the future?
Authors: We have not conducted toxicity studies at this pilot study stage. We plan to conduct basic toxicity studies based on FDA-licensed laboratories only after this structure has been thoroughly tested for antiviral, interferon-inducing, and other types of activity found in dipyridamole.
Reviewer 3 Report
Comments and Suggestions for Authors
The study has been well designed, organized, worked and prepared. The topic is important and actual.
- Corresponding author supposed to be clarify in the author list and noted by asterisk.
- The reaction conditions must be specified in the figure 2. It is also necessary to improve and clarify the visualization of the desired compounds in the scheme (Figure 2), which exact structures the authors prepared, whether two, three or more structures were obtained as target molecules by this method, and what radicals these molecules contain.
Author Response
REVIEWER 3
- Corresponding author supposed to be clarify in the author list and noted by asterisk.
Authors: Corrections were made according to the reviewer's comments.
- The reaction conditions must be specified in the figure 2. It is also necessary to improve and clarify the visualization of the desired compounds in the scheme (Figure 2), which exact structures the authors prepared, whether two, three or more structures were obtained as target molecules by this method, and what radicals these molecules contain.
Authors: This scheme is written for a dynamic structure based on the rules for a Markush formula. This formula is a mixture of derivatives, where each radical can be either hydrogen or a residue of succinic acid or acetic acid. According to the reviewer's comments, the reaction conditions were added to the figure caption: The reaction was carried out in a mixture of dioxane and acetic acid, and the reaction mixture was boiled with a reflux condenser for 25 min.
Round 2
Reviewer 1 Report
Comments and Suggestions for Authors
The authors have addressed all my concerns and suggestions.
Reviewer 3 Report
Comments and Suggestions for Authors
Thanks for getting back to us so soon. I think that this manuscript can be accepted for publication.